# A Descriptive Study of the Carpal Joint of Healthy Donkeys Using Ultrasonography, Computed Tomography, and Magnetic Resonance Imaging

**DOI:** 10.3390/vetsci9050249

**Published:** 2022-05-23

**Authors:** Mohamed Salem, El-Sayed El-Shafaey, Alshimaa M. M. Farag, Sabry El-khodery, Zakriya Al Mohamad, Marwa Abass

**Affiliations:** 1Department of Surgery, Anesthesiology, and Radiology, Faculty of Veterinary Medicine, Mansoura University, Mansoura 35516, Egypt; gamalm@mans.edu.eg (M.S.); eelshafay@mans.edu.eg (E.-S.E.-S.); 2Department of Veterinary Surgery, Salam Veterinary Group, Qassim 51452, Saudi Arabia; 3Department of Internal Medicine, Infectious and Fish Diseases, Faculty of Veterinary Medicine, Mansoura University, Mansoura 35516, Egypt; dr_alshimaafarag@mans.edu.eg (A.M.M.F.); khodery@mans.edu.eg (S.E.-k.); 4Department of Clinical Sciences, College of Veterinary Medicine, King Faisal University, Al Ahsa 31982, Saudi Arabia; zalmohammad@kfu.edu.sa

**Keywords:** carpal, computed tomography, donkey, magnetic resonance imaging, ultrasonography

## Abstract

This study was conducted to establish a detailed anatomic reference for the carpal joint of apparently healthy donkeys using ultrasonography (US), computed tomographic (CT), and magnetic resonance imaging (MRI). Ten orthopedically sound adult donkeys were used for US examination of the carpal joint in each forelimb. Additionally, the carpi of ten donkey cadavers were subjected to CT and MRI examinations. The carpal joint was divided into four zones to simplify examination. US assessment of the carpal joint included transverse and longitudinal sonograms. CT was performed using three planes: axial, sagittal, and coronal. MRI was performed using axial and sagittal planes with two sequences: gradient-echo T1-weighted and proton density. The donkeys’ carpus US, CT, and MRI images were labeled and serially interpreted based on references and anatomical cross-sections. The anatomical characteristics of the carpal joint and the surrounding soft tissue structures were thoroughly described and precisely differentiated on US, CT, and MRI scans. It can be concluded that US, CT, and MRI are effective noninvasive diagnostic imaging tools for evaluating the carpal joint in donkeys. Moreover, these imaging modalities can aid in establishing a reference database for the carpal joint of donkeys, which differs from that of horses.

## 1. Introduction

In quadrupeds, the center of gravity is in the center of the body when the animal is standing and it moves around the center during locomotion [1]. A Stay apparatus allows the animal to stand for an extended period while resting [2]. During steady-state locomotion, the forelimbs bear 60% of the equine’s body weight [3]. Donkeys (*Equus asinus*) are important Egyptian livestock with a significant effect on agriculture, riding tourism, and transportation [4]. Reference data on the musculoskeletal system of donkeys with and without disease are limited [5]. A sound joint is a vital part of properly caring for a donkey’s limbs. The carpal joint is regularly subjected to traumatic affections that induce different lameness grades in donkeys [5].

Carpal lameness in donkeys is challenging to diagnose and necessitates a conjunctive diagnostic protocol of medical investigations and dissimilar symptomatic imaging tools [6,7]. Data on the normal imaging anatomy of the carpal joints of donkeys are essential for understanding carpal affections. Radiography (X-ray), ultrasonography (US), cyphered tomography (CT), magnetic resonance imagery (MRI), and arthroscopy are the most commonly used diagnostic imaging modalities for diagnosing joint lameness in equine [4,6,8].

US is a valuable noninvasive diagnostic tool widely used for examining the carpus, soft tissue, synovia, and articular surfaces in equines [8]. Due to their tomographic nature and high contrast resolution, CT and MRI provide superior diagnostic capabilities over conventional radiography [9]. Furthermore, the imaging of tissue using CT and MRI is done in thin slices, thereby eliminating superimposition, as opposed to radiography, which represents two-dimensional projections of three-dimensional structures [10]. The use of CT and MRI to diagnose lameness in horses has improved the ability to detect soft tissue damage [11]. Clinically, MRI has been used to diagnose intercarpal ligament disorders in humans and to visualize the horse carpal joint’s soft tissue and bone structure [12,13].

In horses, CT has been proven to be a significant imaging mode for accurately identifying carpal lameness undetected by other imaging tools [14,15,16,17]. It provides high-resolution cross-sectional images with spatial separation of structures, allowing for proper visualization of bony structures without superimposition as in plain X-rays [18]. MRI is an efficient and significant imaging tool in equine orthopedics that provides multiplane and multislice cross-sectioned copies of the limbs, specifically soft and hard tissues [19,20]. When the results of other analysis imaging means are inconclusive, MRI is highly recommended to verify the various carpal lameness diagnoses [21,22,23].

In the past few decades, many studies have described the typical imaging features of the carpal region of horses [24,25,26]. However, few studies have focused on donkeys, particularly MRIs of the carpal region. Therefore, this report was designed to define the usual US and CT features of the carpus in donkeys and establish a reference MRI database for donkeys.

## 2. Materials and Methods

### 2.1. Animals

This study involved ten healthy adult donkeys of both sexes with a mean age of 7.3 ± 2.5 years and weighing between 110 and 240 kg. The inclusion criteria for the selected donkeys were that they were deemed (1) clinically healthy, (2) free from any gastrointestinal disorders, (3) free from any evidence of other systemic diseases, (4) easily manageable without any sedation, and (5) they were orthopedically sound with no history of carpal lameness based on clinical and radiographic examinations. The donkeys were purchased from Dakahlia province (Egypt). They were in the stall interior in the animal barn for two weeks before the study.

Donkeys were vaccinated and dewormed on arrival with ivermectin glue (Bimectin^®^, Bimeda Animal Health Ltd., Dublin, Ireland) at a 0.2 mg/kg dose rate. The feeding regimen of the selected donkeys comprised an even and balanced portion consisting of randomly sliced wheat straw, grain (1.5 kg), and crushed corn (1.5 kg), supplemented with all necessary trace elements and minerals, given twice a day at a fixed time (6 am and 6 pm). In addition, tap water was continuously supplied during the day.

Additionally, ten donkey cadavers freshly euthanized for medical reasons unrelated to musculoskeletal system disorders were subjected to CT and MRI examinations of the carpus. The forelimbs were cut at the middle of the radius and amassed immediately after euthanasia, moistened, covered in gauze, closed in plastic bags, and kept at 4 °C for a maximum of 48 h before imaging. Following imaging, the limbs were frozen at −20 °C until sectioning.

For CT and MRI studies, the carpus was thawed to room temperature and washed. The relevant anatomical carpal structures of the donkeys were recognized on US, CT, and MRI images, then categorized and equated with interrelated anatomical references and carpal partitions. An experienced radiologist interpreted all US scans and CT and MRI images.

### 2.2. US Examination

US inspection of the donkeys’ carpi was performed in a standing full-mass-manner situation via a 10 MHz linear probe (CHISON Digital Color Doppler Ultrasound System, ivis 60 EXPERT VET: CHISON Medical Imaging Co., Ltd., Wuxi, China). Following Whitcomb, the carpus was evaluated systematically on longitudinal and transverse planes with 4–6-cm depth for the dorsal, palmar, lateral, and medial surfaces of the carpal joint [27]. For this purpose, the examined donkeys were lightly sedated intravenously (IV), injecting 0.5 mg/kg xylazine HCl (Xylaject 2%; Adwia Co., Cairo, Egypt). Furthermore, the carpus was shaved, cleaned with water, drenched with alcohol, and covered with acoustic pairing gel. Donkeys should be fully weight-bearing during ultrasonographic scanning.

### 2.3. CT Examination

The CT scan of the carpus was performed with a multislice CT machine (Asteion Super 4, Toshiba, Tokyo, Japan) with 120 kV, 160 mA, and 2.0 mm slice width in transverse, sagittal, and 3D scans with bone and soft tissue window image reforms based on [28]. An original scout view was deployed to test for symmetry and to confirm that the entire district was involved in the image. Afterward, transverse (axial), sagittal, and coronal CT scans of each carpus were recreated. In the CT, the carpal joint was divided into four zones. The first zone was situated at the distal end of the radius, the second zone at the first row of carpal bones, the third zone at the second row of carpal bones, and the fourth zone at the proximal end of the third metacarpal bone.

### 2.4. MRI Examination

MRI of the carpus was performed using a low-field magnet (Flexart Toshiba magnet, 0.3 Tesla, Tokyo, Japan) and a human-margin radiofrequency getting coil. A standard scanning protocol was applied using incline-echo T1-bias and proton denseness for the sagittal, transverse, and frontal planes with 4 mm slice thickness, following [29]. During the MRI examination, the carpal joint was divided into three zones. The first zone was situated at the distal end of the radius, the second zone at the first row of carpal bones, and the third zone at the second row of carpal bones. The MRI imaging was performed immediately before the CT scanning. These cadavers’ limbs were imaged within 24 h post-euthanasia.

## 3. Results

### 3.1. US Findings

US images allowed for a conclusive visualization of the soft tissues surrounding the carpus, including the tendon of the extensor carpi radialis tendon (ECRT), the carpal joint capsule (JC), the common digital extensor tendon (CDET), the lateral collateral ligament (LCL), the medial collateral ligament (MCL), the superficial (SDF) and deep digital flexor (DDF) muscles, and the superficial (SDFT) and DDF tendons (DDFT).

The tendon of the extensor carpi radialis muscle appeared as a large, flat, consistent echogenic structure with a linear fiber outline distributed alongside the central dorsal aspect of the carpus in the longitudinal plane and as circular assemblies in the sloping plane.

Meanwhile, the JC revealed a hypoechoic assembly with anechoic synovial fluid behind the echogenic fibers of the ECRT (Figure 1a).

Moreover, the CDET appeared like an elliptical-to-oval homogenous echogenic structure with a thin anechoic sheath around the ECRT and distributed along the dorsolateral aspect of the carpus (Figure 1b).

The LCL appeared as a thin echogenic band that originated from the lateral styloid process of the ulna proximally and was introduced distally into the carpal bones (Figure 2a). The MCL appeared as a thick echogenic structure characterized by a linear fiber pattern at its proximal half and an irregular fiber pattern at the distal half (Figure 2b).

The SDF muscle appeared to be a heterogeneous echogenic assembly with various anechoic and echogenic bands. In contrast, the DDF muscle was only visible as a heterogeneous echogenic structure (Figure 3a). The SDFT and DDFT appeared as echogenic structures with homogeneous linear patterns within the carpal canal parallel to the carpometacarpal combined (Figure 3b).

### 3.2. CT Findings

#### 3.2.1. Sagittal and 3D Scan

The carpal joint consists of two rows of carpal bones forming three articulations: radiocarpal, intercarpal, and carpometacarpal articulations. The first row is made up of three bones (the ulna on the lateral feature, the intermediate in the middle feature, and the radius at the medial surface) (Figure 4).

The second row consists of the first, second, third, and fourth carpal bones from the median to the lateral side. Additionally, the accessory carpal bone appears clear on the 3D scan behind the ulnar carpal bone (Figure 5).

#### 3.2.2. Transverse Scan

In the first zone of the carpal joint, the distal extremity of the radius is the only bone with a clear fossa in the middle of its caudal surface. Moreover, the DDFT and SDFT appeared caudal to the radius. Meanwhile, the second zone started at a point equal to the first row of the carpal joint, which showed the ulna, intermediate, radius, and accessory carpal bones. Moreover, the combined intercarpal was clear and more widely distal. Meanwhile, the third zone at the end of the distal row of the carpal joint displayed the second, third, and fourth carpal bones only. However, the first carpal bone was unclear because it was covered by the second bone in the CT. Distally, at the fourth zone, the proximal margin of the metacarpal bones appeared (Figure 6). Furthermore, the DDFT had an oval-shaped, hypodense structure, whereas the SDFT was hyperdense and smoothly marginated with clearly differentiated borders.

### 3.3. MRI Findings

#### 3.3.1. First Zone

At the CDET groove, the distal extremity of the radius appeared as a low-signal-intensity structure with a thicker cranial cortex. The ECRT had an intermediate signal. Meanwhile, the CDET had a round attendance with moderately high signal strength. The LCL and MCL revealed thin gray bands with intermediate-signal-intensity structures at the lateral and medial aspects of the radius with slightly irregular or ill-defined margins. At this level, the carpal muscles began to be converted into tendons, which caused in areas of both tendons tissues of low signal intensity and muscle tissue of intermediate signal strength. The amount of tendinous tissue increased distally. The SDFT and DDFT appeared as oval intermediate-signal-intensity structures (Figure 7).

#### 3.3.2. Second Zone

In the cross-sectional images, the ulna, intermediate, radius, and accessory carpal bones appeared as black low-signal-strength structures. The ECRT and CDET appeared as an oval black structure with low signal strength. The SDFT and DDFT appeared as oval high-signal-intensity structures. The MCL appeared round near its origin and became more oval-shaped distally. Nevertheless, the LCL appeared narrow proximally and widened distally (Figure 8).

#### 3.3.3. Third Zone

The signal strength was low for the second, third, and fourth carpal bones. The first carpal bone did not appear in this cross-section. All tendinous structures had a strong signal. The signal intensity of the transverse intercarpal ligaments of the carpal bones was intermediate. Two small ligaments were discovered on the palmar side of the carpometacarpal joint, between the second and third carpal bones. The medial ligament began and was inserted between the cannon bone and the base of the second metacarpal bone. Meanwhile, the origin of the lateral intercarpal ligament began between the third and fourth carpal bones and terminated between the third and fourth metacarpal bones (Figure 9).

## 4. Discussion

The distal limb is a common source of forelimb lameness, and the carpus is notoriously underperforming in horses [30]. In Egypt, examination of the musculoskeletal system is limited to conventional radiography (X-ray) and ultrasonography, while computed tomography and MRI are limited to research due to their high financial cost and the unavailability of these devices in veterinary hospitals [31]. The complex anatomy of the carpal joint limits clinical examination and the use of conventional diagnostic imaging techniques, such as X-rays, for diagnosis [29]. To date, this is the first report on the MRI of the carpus and proximal metacarpal region in donkeys. Therefore, this report was designed to provide a full imaging assessment of the carpal region in healthy donkeys through a series of synchronized examinations by US, CT, and MRI.

US is a safe, noninvasive, and practicable diagnostic modality for assessing the soft tissue structures surrounding the carpal multiparty, but limited imaging windows lead to US images with a high tendency for artifacts [32].

In this investigation, a 10 MHz linear probe was used to obtain high-resolution images of the soft tissue structures surrounding the carpal region. This technique is similar to that used in evaluating the carpal joint in horses, as reported previously [33,34].

Ultrasonographic examination of the common digital extensor tendon (CDET) and extensor carpi radialis tendon (ECRT) was easily performed on the dorsal aspect of the carpal joints. Due to their tendonous nature, they appeared as homogeneous echogenic bands, as observed by [34,35]. ECR and CDE tendons were surrounded by the anechoic structure, which represented the tendon sheath, and distributed along the dorsolateral aspect of the carpus, as observed in horses [36,37].

The normal musculature around the carpal region in horses was heterogeneous on the sonogram, with striated and hypoechogenic features [35]. In the current study, the superficial and deep flexor muscle seemed like a heterogeneous echogenic assembly with integrated anechoic and echogenic bands proximal to the accessory carpal joint due to the muscular content, which corresponds to previous findings [27,34]. The SDFT and DDFT distal to the accessory carpal bone appeared as homogenous echogenic bands, which aligns with findings in horses [34,37,38].

Moreover, the JC of the donkey’s carpal joint appeared as a hypoechoic structure with anechoic synovial contents caudal to the ECRT. Meanwhile, the JC appears as an anechogenic layer over the bone in horses, as reported for cattle [27,37,39]. The JC appeared hypoechoic and was below the ECR due to the fat cushion observed at the dorsal aspect of the joint. The longitudinal ultrasonographic scans of the dorsal aspect of the carpus imaged the joint capsules, which were represented by three sacs comprising the radiocarpal, middle carpal, and carpometacarpal joints. The RC sac size was the largest, which is compatible with previous findings on horses [34] and donkeys [18].

In the current study, US did not identify the extensor carpi obliques tendon, which is consistent with previous findings for horses [34] and camels [40]. Furthermore, the LCL and MCL appeared as echogenic structures with thin parallel lines in the longitudinal scan. Similar findings were obtained by [8,41]. The proximal part of the MCL appeared as a thick echogenic structure and was more significant than the LCL one. In contrast, the distal part of the MCL appeared as irregular fibers (short and long ones) due to the different sites of insertion into the carpus bones and metacarpus in the equines. This result was also recorded by [34].

Traditional imaging modalities are in some cases unable to visualize the entire soft tissue and osseous structures of the carpus in donkeys; however, CT can overcome this limitation. CT is an outstanding modality for obtaining tomographic images of the distal limb in equines with a higher spatial resolution and soft tissue difference without the superimposition of the covering structures [14,28]. There is a high degree of similarity between CT images of the carpus region in donkeys and those in horses reported in previous studies, which helps in evaluating abnormal CT scans of the carpus in equines [18,35]. The carpal joint consists of three articulations: the antebrachiocarpal, middle carpal, and carpometacarpal joints [2,42,43].

The acquisition settings used in CT machines are essential for producing an excellent CT image [13,14]. Using a 2.0 mm slice thickness is suitable to gain coronal, sagittal, and transverse plane sections of the carpus region in donkeys. In addition, the use of multislice CT scanners for evaluating the carpal region reduces the scanning time [14,17].

Nonetheless, our findings are relatively consistent compared to the results in the regions assessed in [13,28,42]. The ECR, CDE, and ECO tendons appeared as a bright gray image while the DDFT appeared as a hyperdense, oval structure and the SDFT appeared as hypodense and smoothly marginated with clearly differentiated borders [13].

Small-dimension soft tissue structures like tendon sheaths, bursae, fasciae, and nerves cannot be detected by CT [44], and a lack of fat in the carpus region often makes delineation of flexor and extensor muscles and tendons difficult using CT [18].

The osseous cyst-like lesions observed at the second, third, and fourth carpal bones as oval shapes with regular margins were surrounded by sclerotic rims of different sizes; these findings are consistent with those obtained by [28,45,46]. CT could detect these cysts, but not X-ray imaging due to the increased bone mineral density, as revealed by [47].

MRI is a novel and valuable diagnostic imaging modality for interpreting orthopedic problems in equines. It has numerous advantages over radiography, US, and CT.

MRI provides brilliant noninvasive 3D images of both the soft and hard structures of the carpus [48]. Furthermore, it enables the detection of small and obvious lesions without inducing gross structural variations [19,20]. In this study, the MRI of the carpal joint was performed on isolated limbs of donkeys using a 4 mm slice thickness. There is a slight difference between live and cadaveric carpi; however, this difference does not affect the clinical diagnostic process of carpal lesions. This situation could be attributed to the lack of MRI facilities adjusted for live donkeys in Egypt. Similar recommendations were made by [11,48].

The use of different imaging sequences in MRI allows the identification of physiological variations between normal and abnormal tissues [49,50]. MRI power ranges from the low field (0.2 T) to the high field (1.5 T), which can be used in equine medicine [21,22]. Although high-field MRI is more accurate, both low- and high-field MRI systems provide similar information on abnormal structures [23]. Although high-field MRI is the best method for diagnosing soft tissue lesions within the equine foot, it is not available globally [51]. In this initial study using a low-field MRI system, most structures could be accurately assessed, such as the small palmar ligaments of the carpus, the abaxial margins of the suspensory ligament (SL), the interosseous ligaments between the metacarpal bones, and the carpometacarpal ligaments, which aligns with findings reported in previous studies on horses [21,22,25,52].

Given the scarcity of specific clinical data on ICL [53], our study reported that there were no connections between flexor tendons and ICL, in contrast to the previous studies conducted on horses that found that there were fibrous bundles between the lateral aspects of the SDFT, DDFT and ICL and separated from the flexor retinaculum [25,29]. This finding may be attributed to species variation between donkeys and horses.

The SDFT in this study was like those observed in other prior studies on horses [2,54] and in miniature donkeys [51], appearing as low-intermediate-signal-intensity structures. However, the second accessory ligament of the SDFT was not recorded because this ligament develops a reinforcing ligament, as found in miniature donkeys [51].

Pathological cases are indicated by changes in the signal intensity or normal shape of the anatomical structure in MRI. The tendons and ligaments of the carpal joint in donkeys had distinctive shapes in this study, typically appearing as uniform and intermediate-signal-intensity structures. Similar findings were reported by [22].

In horses, the low signal intensity of the superficial and deep flexor muscles in the proximal aspect of the carpus was reported by [11] due to the variable muscle content in the superficial and deep digital flexor tendons. This is in contrast to our findings, and the inability to detect muscles at this level could be due to species variation.

The medial and lateral collateral ligaments appeared at the levels of the third and fourth carpal bones and had intermediate signal intensity. Most of their fibers were oriented mediolaterally, as described by [15]. Therefore, they could be evaluated as a cross-section. Unlike horses, the fibers were oriented proximodistal [29,55], so dorsal and sagittal sections provided the best evaluation of their fibers.

The intermediate signal intensity bands of the ECR tendon appeared at the first zone level, resulting from an artifact related to lack of tension in the tendon, as explained by [56]. In contrast, the ECR bands at the second zone level appeared with low signal intensity, as also recorded by [11,29,52].

We hypothesized that low field MRI in the current study would not be suitable to evaluate the SL in donkeys. In non-lame horses, the SL appeared to have intermediate to high signal intensity depending on the pulse sequence used, and the intensity of the signal decreased slightly proximally to distally [11,55]. Moreover, previous studies [29,55] have also demonstrated the loss of SL definition during low-field MRI scans, probably because of volume averaging artifacts in the palmar metacarpal vascular structures.

The lack of veterinary imaging laboratories in Egypt with CT and MRI systems modified for live animals was a limitation of this study. As a result, since this aspect of the study was applied to cadaveric limbs, flow and motion artifacts did not occur. Moreover, a small number of healthy donkeys were free from any carpal lesions. Further studies involving several live donkeys with carpal lameness are necessary to identify flow and motion artifacts that often compromise image quality and the use of high-field MRI.

## 5. Conclusions

In conclusion, US, CT, and MRI are helpful noninvasive diagnostic imaging tools for evaluating the carpal joint in donkeys. In addition, these imaging modalities can help in the establishment of a reference for the carpal joint in donkeys, which is different from that in horses, and this could help improve the diagnosis, prognosis, and treatment of carpal lameness in Equidae medicine.

## Figures and Tables

**Figure 1 vetsci-09-00249-f001:**
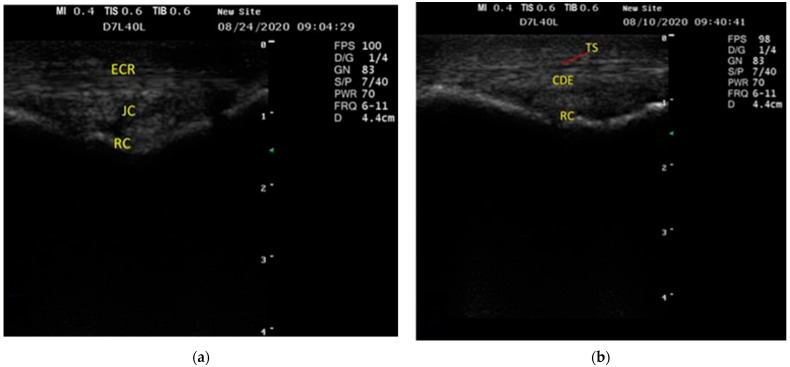
Longitudinal ultrasonographic scan of the dorsal (**a**) and dorsolateral features (**b**) of the carpal joint in donkeys showed that the extensor carpi radialis tendon (ECRT) appeared as a homogenous echogenic structure characterized by a linear fiber outline. Behind the echogenic fibers of the ECRT, the joint capsule appeared as a hypoechoic structure with anechoic synovial fluid. The common digital extensor tendon appeared to be a homogenous echogenic structure with a thin anechoic sheath around the tendon. ECR, extensor carpi radialis tendon; CDET, communal digital extensor tendon; TS, tendon sheath; RC, radiocarpal joint; JC, joint capsule.

**Figure 2 vetsci-09-00249-f002:**
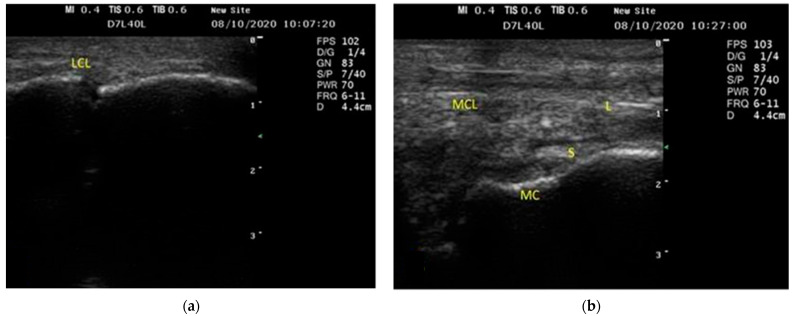
Longitudinal ultrasonographic scan of the lateral (**a**) and medial (**b**) features of the carpal joint in donkeys showed that the lateral collateral ligament (LCL) appears as a thin echogenic band in longitudinal scans. The medial collateral ligament (MCL) seemed like a thick echogenic structure in the US. LCL, lateral collateral ligament; MCL, medial collateral ligament; L, long fibers; S, short fibers; MC, middle carpal bone.

**Figure 3 vetsci-09-00249-f003:**
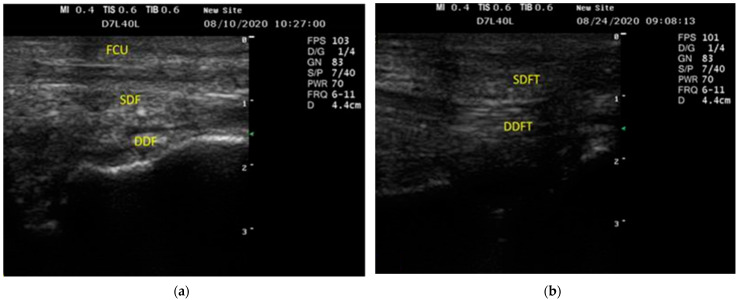
The longitudinal ultrasonographic image of the palmer feature of the carpal multiparty in donkeys showed that the superficial digital flexor muscle seemed to have a heterogeneous echogenic structure with mixed anechoic and echogenic bands. In contrast, the deep digital flexor muscle appeared to be a heterogeneous echogenic structure. (**a**) FCU, flexor carpi ulnaris muscle; SDF, superficial digital flexor muscle; (**b**) DDF, deep digital flexor muscle; SDFT, superficial digital flexor tendon; DDFT, deep digital flexor tendon.

**Figure 4 vetsci-09-00249-f004:**
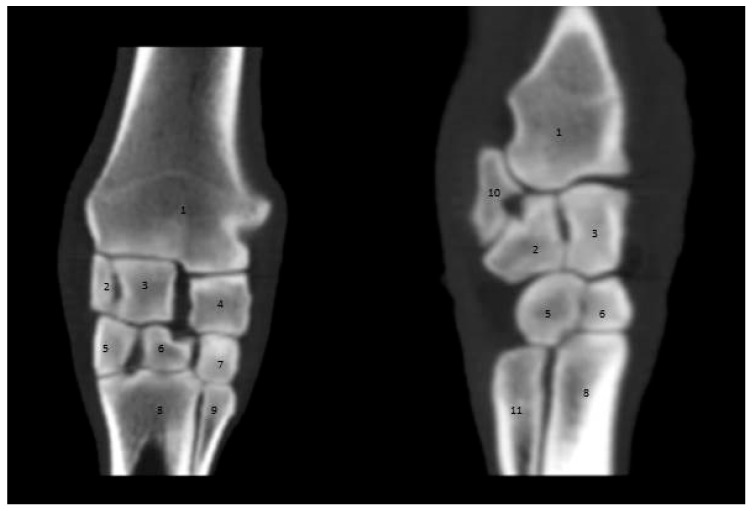
Sagittal computed tomographic scan of the carpal joint in donkeys. (1) Distal radius extremity. (2) Ulnar carpal bone. (3) Intermediate carpal bone. (4) Radiocarpal bone. (5) Fourth carpal bone. (6) Third carpal bone. (7) Second carpal bone. (8) Third metacarpal bone. (9) Second metacarpal bone. (10) Accessory carpal bone. (11) Fourth metacarpal bone.

**Figure 5 vetsci-09-00249-f005:**
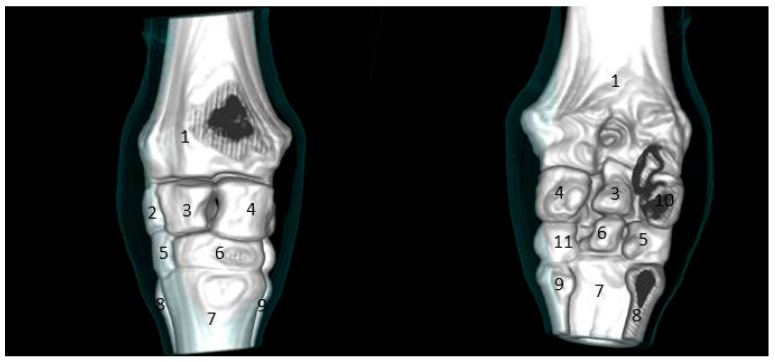
Three-dimensional computed tomographic scan of the carpal joint in donkeys. (1) Distal radius extremity. (2) Ulnar carpal bone. (3) Intermediate carpal bone. (4) Radiocarpal bone. (5) Fourth carpal bone. (6) Third carpal bone. (7) Third metacarpal bone. (8) Fourth metacarpal bone. (9) Second metacarpal bone. (10) Accessory carpal bone. (11) Second carpal bone.

**Figure 6 vetsci-09-00249-f006:**
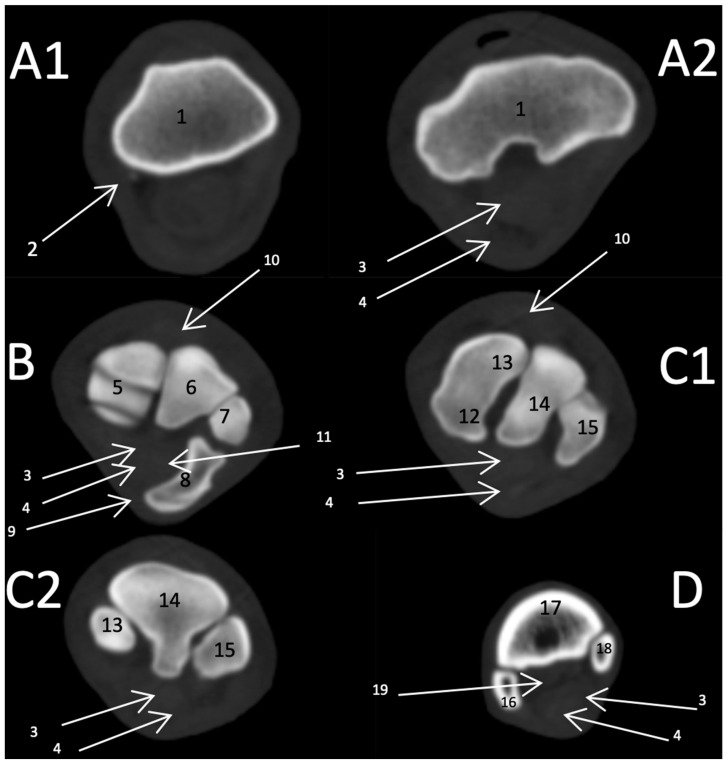
Cross-sectional computed tomographic scan of the carpal joint in donkeys. (**A1**) (1) Radius. (2) Ulna. (**A2**) (3) Deep digital flexor tendon. (4) Superficial digital flexor tendon. (**B**) (5) Radiocarpal bone. (6) Intermediate carpal bone. (7) Ulnar carpal bone. (8) Addition carpal bone. (9) Palmar carpal ligament. (10) Common digital extensor tendon. (11) Carpal canal. (**C1**) (12) First carpal bone. (13) Second carpal bone. (14) Third carpal bone. (15) Fourth carpal bone. (**C2**) (8) Second carpal bone. (9) Third carpal bone. (10) Fourth carpal bone. (**D**) (16) Fourth metacarpal. (17) Third metacarpal bone with the small bone cysts. (18) Second metacarpal bone with the small bone cysts. (19) Suspensory ligament.

**Figure 7 vetsci-09-00249-f007:**
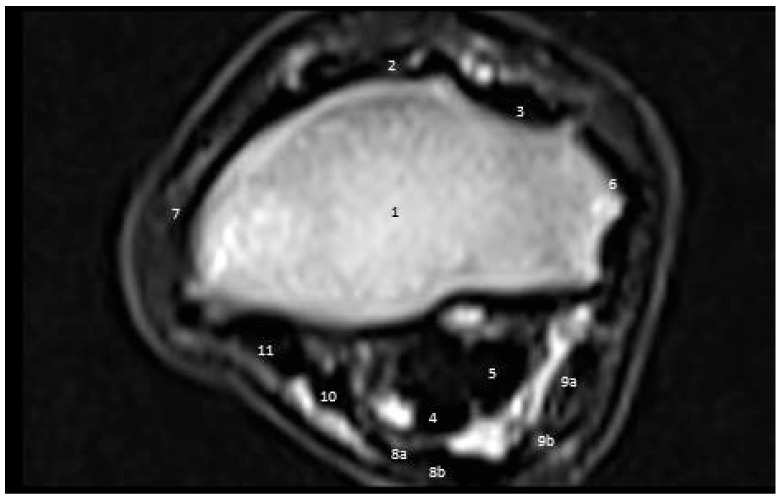
A cross-sectional magnetic resonance imaging scan of the carpal joint in donkeys at the equal of the distal limit of the radius. (1) Radius. (2) Extensor carpi radialis tendon/muscle. (3) Common digital extensor tendon. (4) Superficial digital flexor tendon. (5) Deep digital flexor tendon. (6) Lateral collateral ligament of the carpus. (7) Medial collateral ligament of the carpus. (8) Flexor carpi ulnaris. (8a) Muscle part. (8b) Tendonous part. (9) Ulnaris lateralis tendon. (9a) Muscular part. (9b) Tendon part. (10) Flexor carpi radialis tendon. (11) Carpal fascia.

**Figure 8 vetsci-09-00249-f008:**
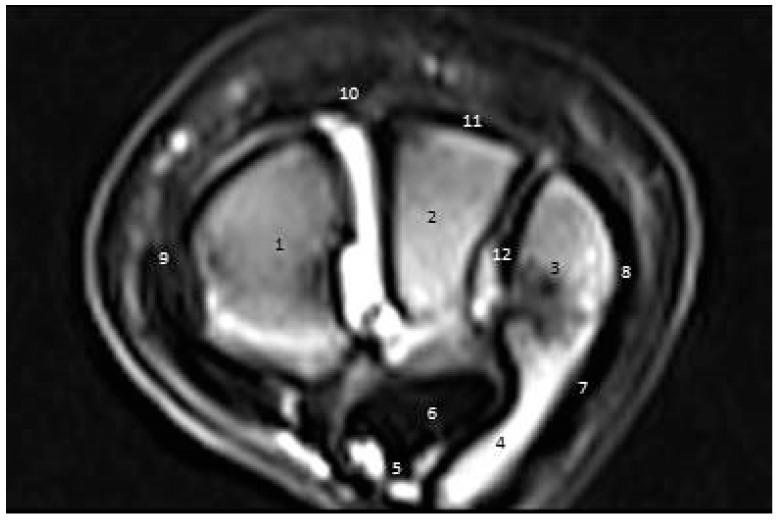
A cross-sectional magnetic resonance imaging scan of the carpal joint in donkeys at the level of the first row of carpal bones. (1) Radiocarpal bone. (2) Intermediate carpal bone. (3) Ulnar carpal bone. (4) Accessory carpal bone. (5) Superficial digital flexor. (6) Deep digital flexor. (7) Ulnaris lateralis tendon. (8) Lateral collateral ligament. (9) Medial collateral ligament. (10) Extensor carpi radialis tendon. (11) Common digital extensor. (12) Transverse intercarpal ligament.

**Figure 9 vetsci-09-00249-f009:**
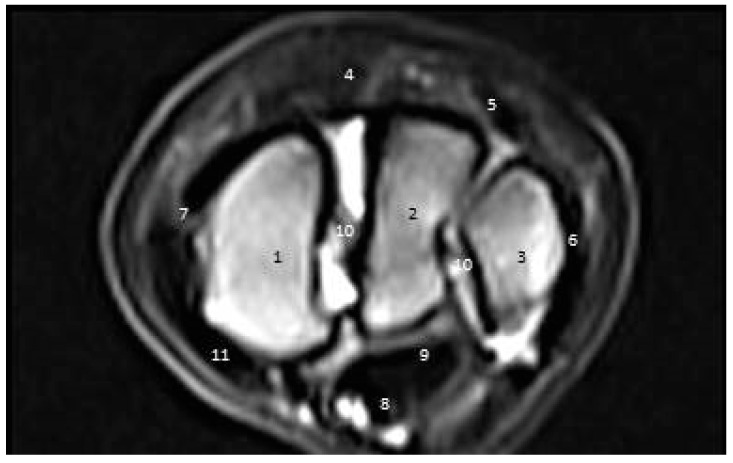
A cross-sectional magnetic resonance imaging scan of the carpal joint in donkeys at the level of the second row of carpal bones. (1) Second carpal bone. (2) Third carpal bone. (3) Fourth carpal bone. (4) Extensor carpi radialis tendon. (5) Mutual digital extensor. (6) Lateral collateral ligament. (7) Medial collateral ligament. (8) Superficial digital flexor tendon. (9) Deep digital flexor tendon. (10) Transverse intercarpal ligaments in the proximal and distal rows of carpal bones. (11) Carpal fascia.

## Data Availability

Not applicable.

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
