# Peer review of "A Descriptive Study of the Carpal Joint of Healthy Donkeys Using Ultrasonography, Computed Tomography, and Magnetic Resonance Imaging"

_vetsci, 2022, doi:10.3390/vetsci9050249_

Round 1
Reviewer 1 Report
Dear authors,
The idea of the manuscript is very interesting. Several comments are mentioned in the attached pdf file. Several English editing is needed.
A comparison with the equine anatomy has to be made in order to give relevance to your work, variations / differences ?? What you mentioned with the intercarpal ligament is not clear. The same with what is mentions in relation to the first carpal bone. Different zones were use for CT and MRI, why not use the same ones and compare both??
Many redundant information about CT and MRI that is not needed.
The ultrasound exam results is very scarce and should be completed.
But I think what could make this paper more valuable is to develop comparison with the horse anatomy.
Author Response
Response to Reviewer 1 Comments
Point 1: The idea of the manuscript is very interesting. Several comments are mentioned in the attached pdf file. Several English editing is needed.
Response 1: Thank you for bringing this to our attention. The English editing of the manuscript has been done, and all language issues have been addressed.
Point 2: A comparison with the equine anatomy has to be made in order to give relevance to your work, variations / differences ??
Response 2: Thank you for bringing this to our attention. All the required changes were done in the text, as recommended. (lines 298-407).
Point 3: What you mentioned with the intercarpal ligament is not clear. The same with what is mentions in relation to the first carpal bone.
Response 3: In contrast to previous studies on horses, which found fibrous bundles between the lateral aspect of SDFT and DDFT and ICL and clearly separated from the flexor retinaculum[1,2], our findings revealed no connections between flexor tendons and ICL, which could be attributed to species variation between donkeys and horses.
Point 4: The same with what is mentions in relation to the first carpal bone.
Response 4: The first carpal bone was unclear because it was covered by the second bone on CT and did not appear in the MRI scan. These findings agree with [29], which recorded that a first carpal bone was present in 8/30 limbs (26.7%) of horses.
Point 5: Different zones were use for CT and MRI, why not use the same ones and compare both??
Response 5: Since the CT scan is used to evaluate the hard tissues, the MRI scan is utilized to evaluate the soft tissues. In the current study, the CT scan was done according to [28], and the MRI scan was performed in accordance with [29]. They divided the carpus region into different zones in sequential dorsal plane images from dorsal to palmar. The first zone is situated at the distal end of the radius, the second zone is located at the first row of carpal bones, the third zone is located at the second row of carpal bones, and the fourth zone is located at the proximal end of the third metacarpal bone.
Point 6: Many redundant information about CT and MRI that is not needed.
Response 6: Thank you for bringing this to our attention. All the required changes were done in the text as recommended throughout the manuscript.
Point 7: The ultrasound exam results is very scarce and should be completed.
Response 7: Thank you for bringing this to our attention. All the required changes were done in the text as recommended. (Lines 296-333)
Point 8: But I think what could make this paper more valuable is to develop comparison with the horse anatomy.
Response 8: Thank you for bringing this to our attention. All the required changes were done in the text as recommended in decisions section.
Point 9: Does the introduction provide sufficient background and include all relevant references? "Must be improved"
Response 9: Thank you for bringing this to our attention. All the required changes were done in the text as recommended in introduction section.
Point 10: Are all the cited references relevant to the research? "Can be improved"
Response 10: Thank you for bringing this to our attention. All the required changes were done in the text as recommended.
Point 11: Are the results clearly presented? "Must be improved"
Response 11: Thank you for bringing this to our attention. All the required changes were done in the text as recommended in the results section.
Point 12: Are the conclusions supported by the results? "Must be improved"
Response 12: Thank you for bringing this to our attention. All the required changes were done in the text as recommended in the conclusion section.
Reviewer 2 Report
The authors studied the carpal joints of donkeys using CT, US and MRI and found these approaches helpful for assessment of joint status.
Comments
- All the typos should be corrected.
- The authors should discuss which approach is prevalent for veterinary imaging of donkeys and which can be used predominantly in veterinary practice.
Author Response
Response to Reviewer 2 Comments
Point 1: All the typos should be corrected.
Response 1: Thank you for bringing this to our attention. The English editing of the manuscript has been done, and all language issues have been addressed.
Point 2: The authors should discuss which approach is prevalent for veterinary imaging of donkeys and which can be used predominantly in veterinary practice.
Response 2: Thank you for bringing this to our attention (Lines 281-283 ).
Round 2
Reviewer 1 Report
Suggested changes had been made.